# Advanced Robotic System with Keypoint Extraction and YOLOv5 Object Detection Algorithm for Precise Livestock Monitoring

**Balaji Natesan** [1], **Chuan-Ming Liu** [2,*], **Van-Dai Ta** [3] and **Raymond Liao** [4]

1    International Graduate Program of College of Electrical Engineering and Computer Science, National Taipei University of Technology, Taipei City 106, Taiwan; t109999405@ntut.edu.tw

2    Department of Computer Science and Information Engineering, National Taipei University of Technology, Taipei City 106, Taiwan

3    Samsung Display Vietnam (SDV), Yen Phong Industrial Park, Bac Ninh 16000, Vietnam; t104999002@ntut.edu.tw

4    Yuanshang Technology Co., Ltd., New Taipei 242, Taiwan; raymond.liao@yst93.com

*    Correspondence: cmliu@ntut.edu.tw; Tel.: +886-2-2771-2171

**Abstract:** Molting is an essential operation in the life of every lobster, and observing this process will help us to assist lobsters in their recovery. However, traditional observation consumes a significant amount of time and labor. This study aims to develop an autonomous AI-based robot monitoring system to detect molt. In this study, we used an optimized Yolov5s algorithm and DeepLabCut tool to analyze and detect all six molting phases such as S1 (normal), S2 (stress), S3–S5 (molt), and S6 (exoskeleton). We constructed the proposed optimized Yolov5s algorithm to analyze the frequency of posture change between S1 (normal) and S2 (stress). During this stage, if the lobster stays stressed for 80% of the past 6 h, the system will assign the keypoint from the DeepLabCut tool to the lobster hip. The process primarily concentrates on the S3–S5 stage to identify the variation in the hatching spot. At the end of this process, the system will re-import the optimized Yolov5s to detect the presence of an independent shell, S6, inside the tank. The optimized Yolov5s embedded a Convolutional Block Attention Module into the backbone network to improve the feature extraction capability of the model, which has been evaluated by evaluation metrics, comparison studies, and IoU comparisons between Yolo's to understand the network's performance. Additionally, we conducted experiments to measure the accuracy of the DeepLabCut Tool's detections.

**Keywords:** aquaculture; Yolo; lobster; robotics; Yolov5s-CBAM; keypoint detection; computer vision

**Key Contribution:** In this study, we proposed an integrated system combined with robotics and deep neural networks to assist in the analysis and detection of the six molting phases of lobster. Such an integrated system opted with an optimized Yolov5s algorithm and DeepLabCut Tool to significantly improve upon labor-intensive traditional methods.

## 1. Introduction

In 2021, the value of the lobster market in the seafood industry has been raised to USD 6.3 billion, with experts predicting this market value to reach up to USD 11.1 billion by 2027. This data proves industries grow significantly. According to a survey in 2022–2023, the global fishing market gross value reached up to USD 3.63 billion [1]. The countries in the Indo-Pacific region, such as Vietnam, China, and Indonesia, as shown in Figure 1, have seen some impressive growth in their lobster businesses lately [2]. These developments indicate a steady yearly rise in the value of seafood consumed by people [3]. However, every year seafood industry loses millions of dollars due to insufficient monitoring of their livestock. Through manual monitoring, there is a significant boost of up to 70% in the survival rates of

sea life, but it is time-consuming and labor-intensive [4]. During the COVID-19 lockdown, the seafood industry faced a higher rate of livestock deaths, largely due to a reduction in manual monitoring [5]. This highlights the significance of autonomous monitoring systems in managing livestock. In recent years, researchers around the world have proven that using autonomous monitoring systems is a helpful method to increase the survival rate of animals and enhance productivity by effectively controlling diseases and automatically detecting abnormal moments in lobsters without human intervention [6].

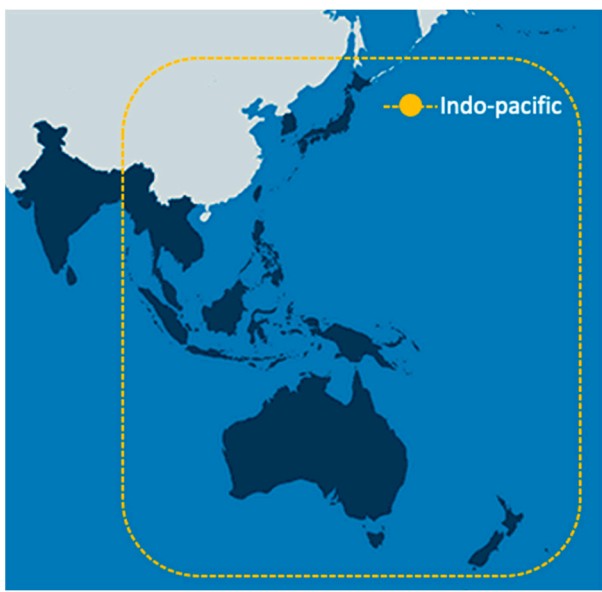

**Figure 1.** Image of Indo-Pacific lobster territory.

Deep neural networks (DL) are increasingly being used in monitoring systems for various industries, including aquaculture disease detection, health routines, and so on [7]. In 2019, Rauf et al., [8] developed a 32-deep layer CNN architecture for a decision-making system for recognizing fish species, outperforming existing methods. In 2021, an author [9] developed a deep CNN model that accurately classifies feeding and non-feeding fish behavior into four levels—none, weak, medium, and strong—achieving an impressive accuracy rate of 90%. In the same year, ref. [10] proposed a method using a CNN and LSTM architecture to accurately evaluate feeding and non-feeding behavior with an 80% accuracy rate. Also, in a separate study in 2021, an author analyzed fish feeding behavior to promote healthy practices using a support vector machine (SVM) and gray gradient symbiosis matrix, with preprocessing techniques like image enhancement, background subtraction, and target extraction to enhance model performance [11]. Moving to 2022, Wang et al., proposed an intelligent feeding decision system using the Appearance-Motion Autoencoder Network (AMA-Net), a semi-supervised learning approach that allows for accurate fish feeding measurements [12]. Also, in 2022, Ahmed et al., [13] developed a deep learning technique for early salmon fish illness detection using machine learning and image processing techniques. They segmented fish photos using K-means clustering to extract essential features and trained a SVM model to identify healthy or diseased fish. The model achieved high accuracy rates of 91.42% and 94.12%, both with and without picture augmentation techniques. This study highlights the potential of machine learning in improving aquaculture sustainability through smart applications in monitoring systems.

Even though multiple studies discuss the monitoring system, there still exists a gap between the ground truth and the predicted images, which is evident in the accuracy results. In 2016, ref. [14] proposed a novel and innovative model named You Only Look Once (YOLO), which is renowned for its IOU-based detection and the network consisting of a convolutional backbone and multiple detection heads, enabling real-time and accurate object detection tasks. In 2023, an author [15] introduced a novel method based on the Yolov5

model that incorporates ghost convolutions as replacements for standard convolutions, achieving an impressive accuracy of 98.139% in detecting white shrimp surfacing. Also in 2023, another author [16] used Yolov5s-CBAM to accurately detect a mushroom-growing house environment, achieving a 98% accuracy rate. The lightweight model, designed for lower-end devices, enabled additional feature extraction operations. However, tracking animals remains a challenging issue.

To provide a brief overview of the DeepLabCut tool in 2022, Lauer et al., proposed an efficient and challenging computer vision technology that can identify the poses of multiple animals using keypoints. The DeepLabCut tool utilizes a multi-fusion architecture and multi-stage decoder to process images and identify keypoints in animals through feature detection [17]. Fujimori et al., [18] proposed a method using the DeepLabCut tool to track and classify animals' behavior using skeletal information. They monitored static behaviors like sitting, lying down, and eating and dynamic behaviors like walking and jumping. Following that in 2023, an author [19] introduced a method using the DeepLabCut tool (version 2.2) for tracking small-bodied fish, and this method has proven effective in enhancing fish passage management and studying behavior through keypoint detection. This breakthrough in the monitoring industry allows for the tracking of movements of small animals, including legs, limbs, heads, and the entire body, thereby enhancing our understanding of animal behavior.

Despite the widespread use of AI technology in animal behavior monitoring, researchers have not conducted research on detecting lobster molting. Molt refers to the cyclical patterns in which lobsters prepare for molting and then recover [20]. Molting is a crucial part of the lobster life cycle, and male lobsters molt more frequently per year compared to females. After molting, lobsters require close monitoring to ensure they have enough recovery food, as this process increases stress levels and reduces survival rates [21]. This paper employs techniques to identify lobster molting, a process crucial for their growth and overall health. Also in this paper, we present the results and experiments conducted on three major lobster species in the Indo-Pacific region: *Panulirus homarus*, *Panulirus longlines*, and *Panulirus ornatus*.

The following key findings and insights were obtained from the testing results and discussion:

1. Novel Backbone Architecture with Attention Mechanism: We have utilized a novel backbone architecture that combines the strengths for extracting the rich features from the input data and focuses specifically on the lobster, enhancing the accuracy and performance of the molting.
2. Utilization of Keypoint Detection: To accurately identify the lobster molting, we have incorporated keypoint detection. This technique further improves the effectiveness of the proposed architecture by capturing the keypoints in the lobster's body.
3. Analysis of Posture Change Frequency: As part of the study, we compared the frequency of posture change exhibited by lobsters in the molting period. The results offer important new understandings of the movements and behavioral characteristics of lobsters throughout the molting process.

## 2. Materials and Methods

### 2.1. Data Collection

The dataset used in this study was collected from three species of lobsters: *P. homarus*, *P. longipes*, and *P. ornatus*. We focused on *P. homarus* because it is the most common species. We used a FLIR IR camera with a resolution of 1080 × 1920 to collect 1000 images of lobsters. We divided the images into an 8:2 ratio for training and testing. The training set consists of 800 images made up of a specific allocation, with 300 images for each normal and exoskeleton posture. Given a small postural similarity between the exoskeleton and normal states, this allocation aided the model's successful learning of differentiating traits. Additionally, 200 images were allocated for stress posture. To assess the model's performance, we utilized unseen data during the testing phase, which featured distinct

environments and background colors. To ensure the model's stability, we captured training and testing images under varying lighting conditions. We employed an open-source tool called Roboflow for labeling the data.

### 2.2. Convolutional Block Attention Module Fused in Yolov5s

Yolo is a widely used object detection algorithm known for its high real-time accuracy and its ability to be optimized for specific use cases. Yolo works by predicting the intersection area between the predicted bounding box and the actual ground-truth bounding box. This allows us to evaluate the IOU (intersection over union) metric and the accuracy of the predictions. The Yolov5 consists of three important models: Yolov5s, Yolov5m, and Yolov5l. Each model differs in the number of layers, filters, and network depth. Due to these factors, the performance and weight of the model differ. The Yolov5s (small): the smallest and fastest version. Yolov5m (medium): a balance between speed and accuracy. Yolov5l (large): larger and more accurate than medium. Due to the computational power factor in our embedded system, we utilized a lightweight model called Yolov5s. Yolov5s is a popular choice because it is efficient and lightweight. However, we found that Yolov5s had some limitations in accurately detecting lobsters in underwater conditions. To address these limitations, we adopted Yolov5s-CBAM, which is an enhanced version of Yolov5s that incorporates the Channel Attention Mechanism (CBAM). CBAM helps to improve the accuracy of Yolov5s by focusing on the most important channels in the input image. With Yolov5-CBAM, we were able to successfully identify and distinguish two lobster postures—s1 (normal) and s2 (stress)—and dead skin s6 (exoskeleton). The Yolov5s model comprises four essential modules, as depicted in Figure 2: the input module, the backbone module, the neck module, and the prediction module.

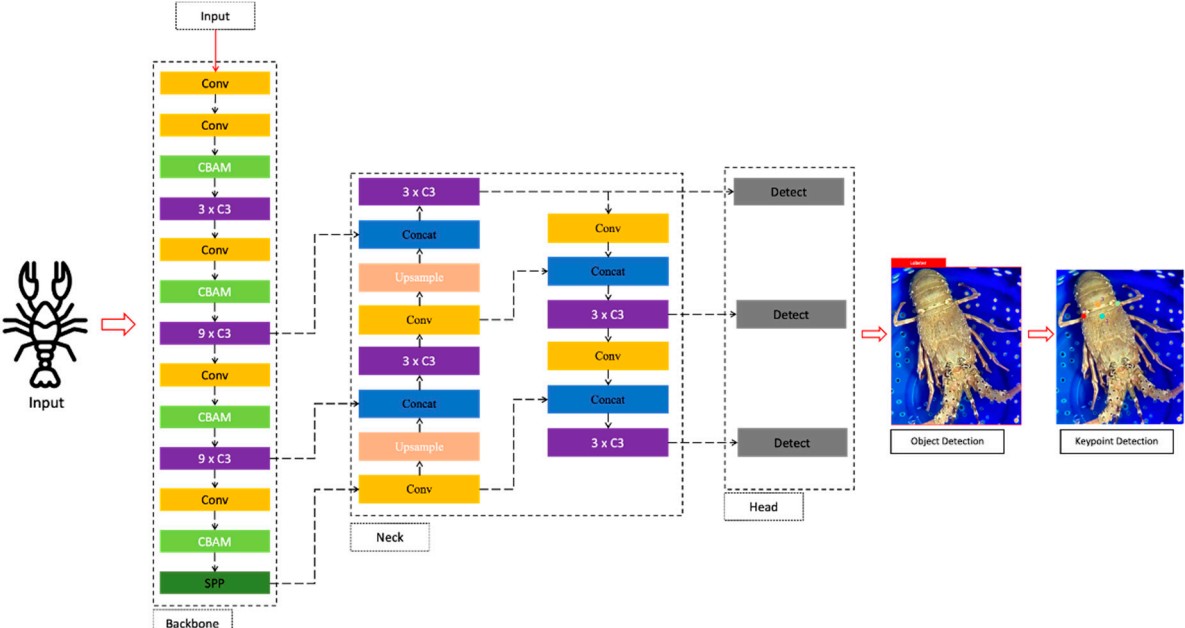

**Figure 2.** Yolov5s-CBAM architecture.

To enhance the performance of the model, we incorporate the Convolutional Block Attention Module (CBAM) within the backbone module. The primary purpose of the backbone module is to extract crucial features from the input data. This module consists of five convolutional layers, strategically designed to gather informative features from the input images. To enhance feature processing between convolutional layers, we introduced CSP (Cross-stage Partial) blocks. These CSP blocks facilitate the extraction of the most pertinent features from the convolutional layer. To further optimize the model's performance, we integrated the Convolutional Block Attention Module between these CSP blocks, as

depicted in Figure 2. This module effectively enhances the model's attention to significant features during processing. Finally, to extract features at various scales and resolutions, we utilized the Spatial Pyramid Pooling (SPP) layer.

$$M_c(F) = \sigma((MLP(AvgPool(F))) + (MLP(MaxPool(F)))) \tag{1}$$

The mentioned Convolutional Block Attention Module combined with channel attention and spatial attention to capture the important features F in the dimension of channel axis and spatial axis as shown in Figure 3. The network integrated with the attention mechanism can effectively use the target object regions and capture features from this information, resulting in improved performance. The channel attention mechanism learns to weigh the importance of feature maps across channels as shown in Figure 4a. Channel attention uses a multilayer perceptron (*MLP*) with a hidden layer, as explained in Equation (1).

$$M_s(F) = \sigma\left(f^{7 \times 7}\left(\left[F_{avg}^s; F_{max}^s\right]\right)\right) \tag{2}$$

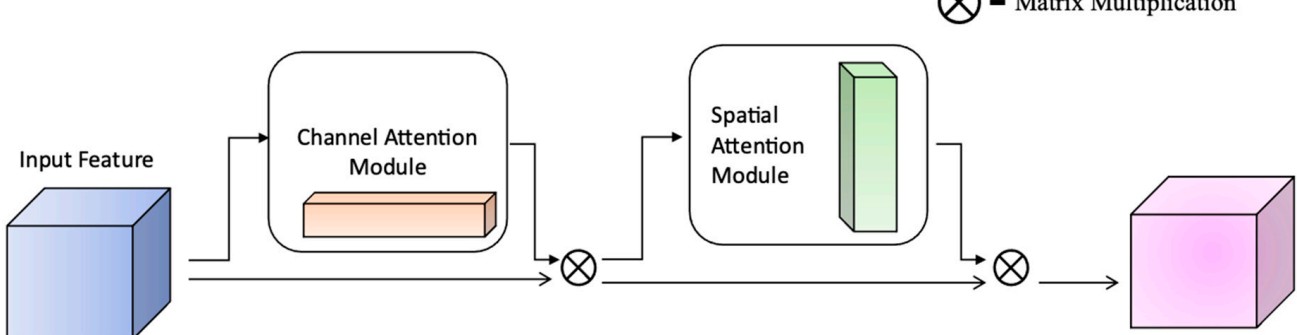

**Figure 3.** CBAM (Convolutional Block Attention Module) structure.

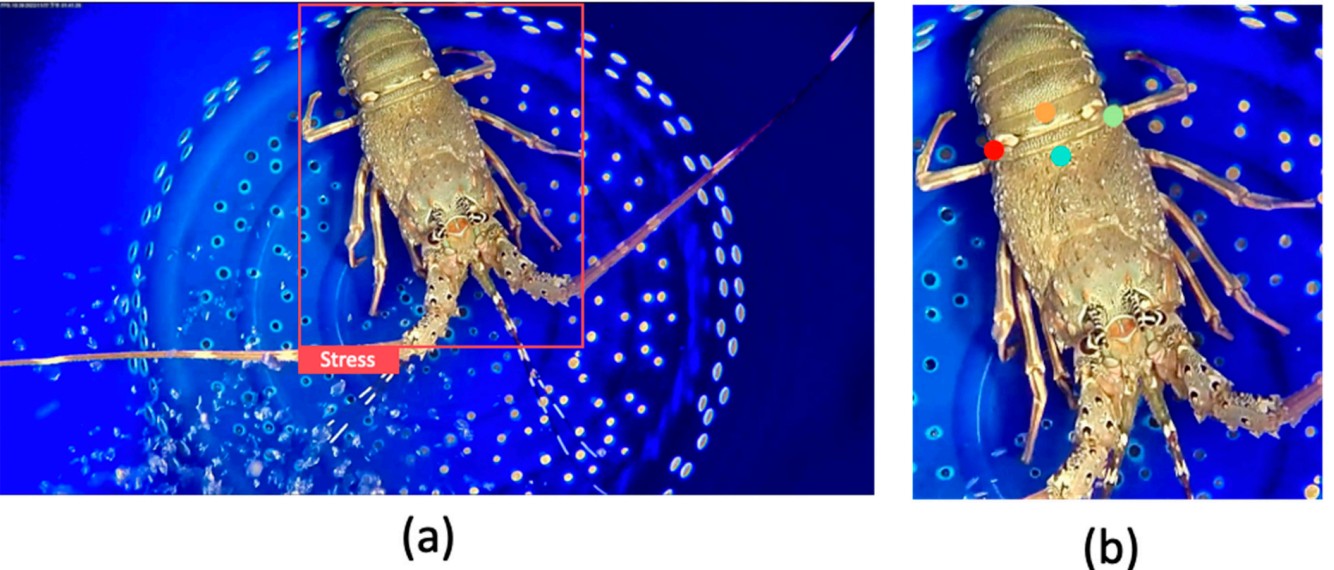

**Figure 4.** Lobster stress posture detection (**a**); lobster with keypoint detection after preprocessing the image (**b**), where cyan represents x1, orange x2, light green y1, and red y2.

The spatial attention mechanism learns to weigh the importance of features within each feature map. Spatial attention also uses the average-pooling and max-pooling operations, as explained in Equation (2).

The neck module, $N_m$, proceeds to concatenate the spatial attention features, C, with the original features. The earlier neck modules will do an up-and-down sampling block.

By incorporating all the features, the prediction module $P_m$, ultimately, estimates the coordinates, position, and species.

$$m(x, y) = \max(m_n(x, y)) \tag{3}$$

To minimize the noise for efficient keypoint detection, we generate a binary mask m for the predicted object, as described in Equation (3), where $x$ and $y$ denote the position of the mask.

$$f_n(x, y) = (m(x, y)) * (I_m(x, y)) \tag{4}$$

To obtain the final masked image, $f_n$, we multiply the binary mask with the input image, $I_m$, according to Equation (4), which will transfer the detected Figure 4a,b as input for keypoint detection to reduce the noises.

### 2.3. Keypoint Detection

For keypoint detection, DeepLabCut Tool is an open-source system that focuses on estimating the pose of a single target using Keypoint, which is more popular for use on smaller animals. DeepLabCut Tool uses a transfer learning method to identify the keypoint with less dataset [19]. The keypoint has been applied for S3, S4, and S5. The backbone of the network are pre-trained network such as Resnet 50 with 2048 channels. DeepLabCut restricts Resnet50 network downsampling to only 16 times to handle the larger size of the feature map size. DeepLabCut uses the heat map to identify the location of keypoints. DeepLabCut has two loss functions, such as binary cross entropy loss and Huber loss in training [17].

$$y = a(W(f_n) + b) \tag{5}$$

$y$ is the predicted output of the neural network, $a$ is an activation function that applies a non-linear transformation to the weighted input. $W$ is the weight of neural networks, which are learned parameters during training. $f_n$ is the input image for the DeepLabCut. $b$ is the bias term which is a constant value added to the weighted input to control the output in Equation (5). We use this architecture to identify the lobster molting. The keypoints give leverage to identify the shredding by calculating the moving frame $x$, $y$ keypoint.

$$d = sqrt\left(\left(x2 - x1\right)^2 + \left(y2 - y1\right)^2\right) \tag{6}$$

In this Equation (6), $d$ represents the Euclidean distance between the keypoints, and cyan represents $x1$, orange represents the $x2$, and light green represents the $y1$, red represents the $y2$ which are the four coordinates keypoints, as shown in Figure 4b. The $x1$ and $x2$ will provide a higher variance than $y1$ and $y2$ while molting. By including this calculation, we can gain a better understanding of the lobster's dimensions and we can calculate the distance.

## 3. Discussion

### 3.1. Implementation

Pytorch was used for Yolov5s-CBAM implementation, while TensorFlow was used for keypoint detection, both models were developed using Python. We have chosen the Robot Operating System for the robot control. During the training phase, we utilized a batch size of 64 for Yolov5s-CBAM. To accelerate the training, an RTX 3070-ti GPU was utilized. The training of the model spanned 140 epochs, a learning rate of 0.001, and a dropout rate of 0.4, and we meticulously fine-tuned the hyperparameters throughout the training process to optimize performance. For the robot, we equipped it with a mini-computer featuring an Intel i5 processor, empowering it to execute the entire monitoring process.

### 3.2. Behavior Analysis

With the valuable insights provided by experts in the field, we discovered an essential pattern in lobster behavior, as shown in Figure 5. The molting process in lobsters consists

of five significant phases—s1 (Normal), s2 (Stress), and s3–s5 (Molt)—with the final phase involving exoskeleton shedding s6, which is not considered a posture. Lobsters typically maintain a stressed posture for an extended period before molting. The duration of a stage was accurately determined by continuously monitoring seven molting lobsters using a robot for six hours. We recorded the frequency of posture changes between normal and stressed positions during this period. Throughout the observation period, the lobsters predominantly stayed in the stress posture, accounting for approximately 80% of the total time. On the other hand, they exhibited normal posture only about 20% of the time.

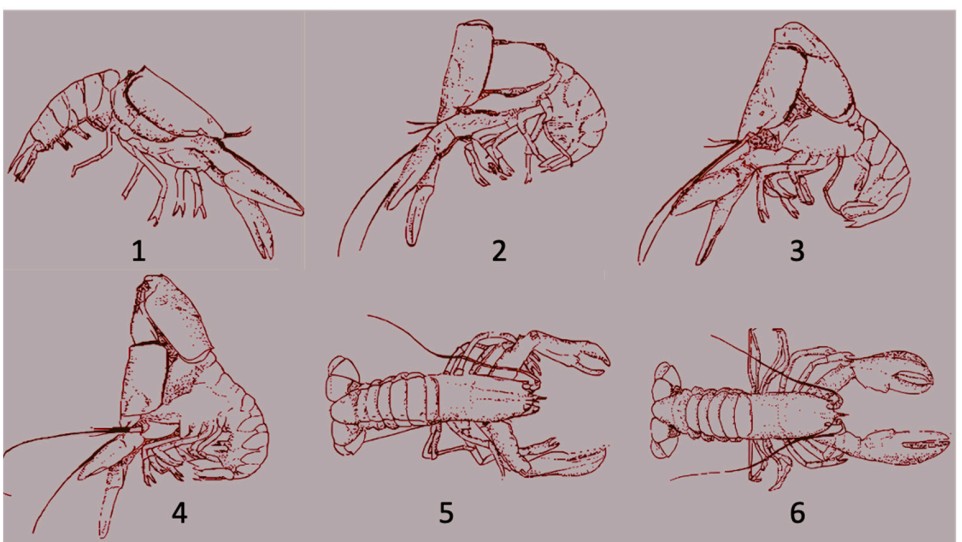

**Figure 5.** Distress postures (1,2); molting stages (beginning, middle, end) (3,4); exoskeleton (5); normal postures (6).

### 3.3. System Design

The designed robot system features four motors that enable it to move the FLIR camera across the 16 lobster boxes, as illustrated in Figure 6. The system was enabled with ROS for robot arm and movement controls. The x, y, and z coordinates for each lobster box are stored in a ROS topic. The system subscribes to this topic and uses the values to position the motors. The robot arm's lateral movement is controlled by x and y coordinates, enabling container transition, while its vertical motion is governed by the z-axis. The system has labels for each tank, and when it moves to another container, it automatically updates its information to the new lobster.

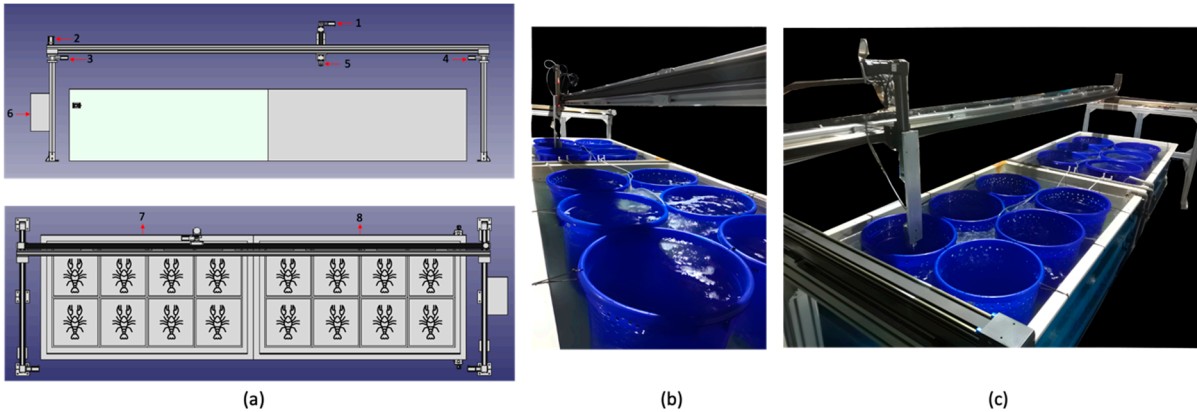

**Figure 6.** Lobster monitoring setup. (**a**) Components: *z*-axis camera motor (1); x and y-axes motors (2,3,4); IR camera (5); CPU cabinate (6); individual tanks (7,8); (**b**) real-world configuration; (**c**) camera placement inside the container.

The robot system was powered by two deep neural networks, Yolov5s-CBAM and DeepLabCut. Upon arrival at the container, the robot will begin analyzing the current lobster's past and present posture data. The analysis begins by observing the transition between a lobster's normal and stressed postures using Yolov5s-CBAM. The system detects keypoints around the hip area of a lobster when it remains in a stressed posture for 80% of the data period.

Figure 7 shows the lateral view of the real-time variance that will occur during keypoint detection. The system looks for substantial variance in $x1$ and $x2$, during the molting process. After the molting, the lobster will not show significant variance in the keypoint measure. The system will automatically reimport Yolov5s-CBAM to distinguish between lobster and exoskeleton, as illustrated in Figure 8.

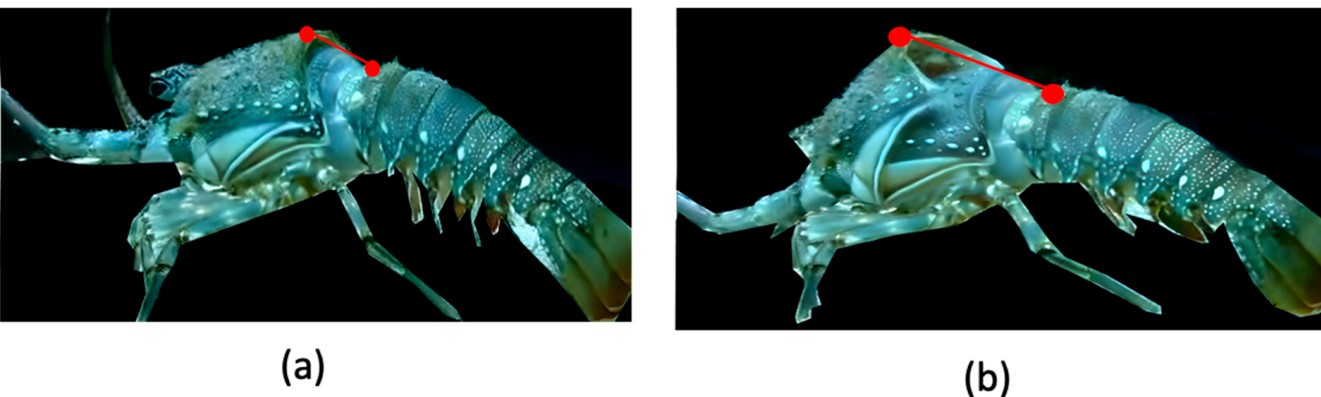

**Figure 7.** Lateral view of lobster: starting stage of molting (**a**); mid-stage of molting (**b**).

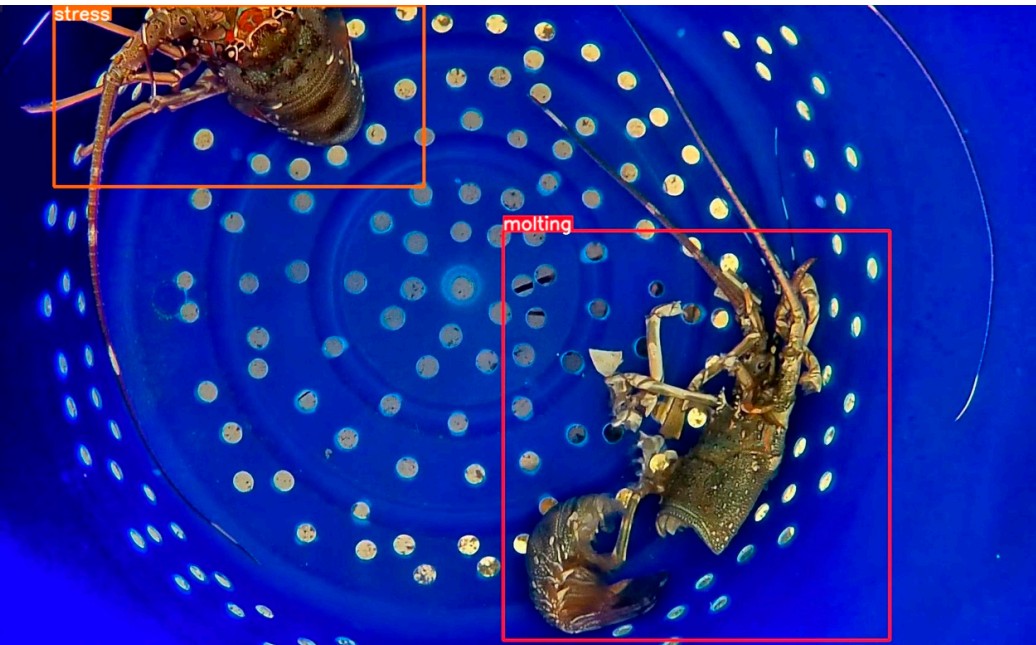

**Figure 8.** Yolov5s-CBAM detection results between lobster and exoskeleton post molt.

The monitoring system effectively detects molting events and takes necessary actions to ensure lobster health in the aquaculture environment by integrating these functionalities. The system flowchart is mentioned in Figure 9.

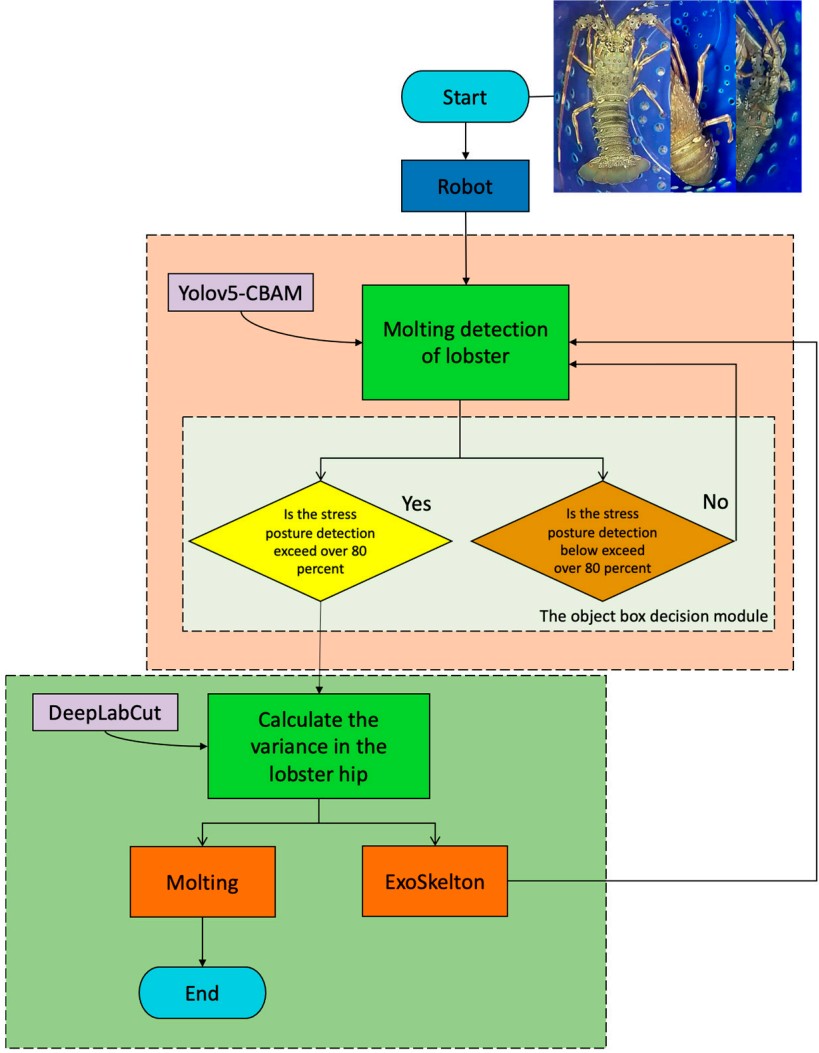

**Figure 9.** Flowchart.

### 3.4. Experimental Results

The Yolov5 model is gaining popularity among researchers and practitioners due to its efficient and effective solution for object detection tasks. Yolov5s, in particular, is renowned for being a lightweight model, making it more suitable for deployment on lower-end devices. The model's performance is evaluated using metrics like precision, recall, mAP (Mean Average Precision), and F1-score, as detailed in Equations (7)–(10). TP stands for True Positive, TN for True Negative, FP for False Positive, and FN for False Negative. The preliminary setup involves using the base Yolov5s without any additional operations. These evaluation metrics help verify the efficiency and effectiveness of the Yolov5s model.

$$Precision = \frac{(\text{TP})}{(\text{TP} + \text{FP})} \tag{7}$$

$$Recall = \frac{(\text{TP})}{(\text{TP} + \text{FN})} \tag{8}$$

$$F1_{Score} = \frac{(2 \times \text{TP})}{(2 \times \text{TP} + \text{FN} + \text{FP})} \tag{9}$$

$$mAP = \frac{1}{|Q_R|} \sum_{q=Q_R} AP(q) \tag{10}$$

The initial experiment assessed the effectiveness of various Yolov5 models, including Yolov5s, Yolov5m, and Yolov5l, in identifying the most suitable option. The results showed that Yolov5s performed better in terms of recall and F1-score. Yolov5m showed the best precision but slightly lower recall and mAP, while Yolov5l showed lower recall but improved precision and mAP results, as indicated in Table 1. The Yolov5m and Yolov5l models demonstrated promising performance, but their larger model sizes posed challenges, particularly for our CPU. Due to their higher computational demands, we were unable to continue testing these models. The evaluation results indicate that Yolov5s is the most suitable model due to its balance between performance and size, making it the preferred choice for our setup.

**Table 1.** Comparison between Yolov5s, Yolov5m, and Yolov5l.

| Model | Size (MB) | Precision | Recall | mAP@0.5 | F1-Score | Parameters |
|-------|-----------|-----------|--------|---------|----------|------------|
| Yolov5s | 14.4 | 96.2 | 96.9 | 95.9 | 95.8 | $7.03 \times 10^6$ |
| Yolov5m | 42.2 | 96.8 | 95.7 | 94.2 | 93.4 | $2.09 \times 10^7$ |
| Yolov5l | 92.8 | 98.1 | 96.9 | 98.7 | 95.6 | $4.61 \times 10^7$ |

To improve Yolov5's performance by introducing a more advanced feature extraction tool to support the network. The Yolov5s-SE and Yolov5-BI-FPN models are evaluated for their exceptional feature extraction skills, as well as the Yolov5s-CBAM. In particular, Yolov5 SE is the effect of spatial attention mechanisms on how well the model manages object context and fine-grained features [16]. On the other hand, Bi-FPN (Bi-Directional Feature Pyramid Network) was chosen for Yolov5s with the aim of improving object identification and feature integration at various scales and increasing model accuracy [22]. Throughout this study, we observed that Yolov5s-CBAM exhibited better precision, mAP, and F1-score, indicating enhanced overall performance. Yolov5s-SE demonstrated superior recall in object detection, as shown in Table 2. Yolov5s-BI-FPN's performance was slightly lower than other models due to limited dataset availability, impacting its training and evaluation. The evaluation results indicate that the CBAM module significantly enhances object detection, particularly with fewer images. The CBAM module's potential to enhance Yolov5's detection capabilities, resulting in improved precision and F1-score, makes it a valuable addition for specific use cases.

**Table 2.** Comparison between Yolov5s, Yolov5s-Bi-FPN, Yolov5-SE, and Yolov5s-CBAM.

| Model | Size (MB) | Precision | Recall | mAP@0.5 | F1-Score | Parameters |
|-------|-----------|-----------|--------|---------|----------|------------|
| Yolov5s | 14.4 | 96.2 | 96.9 | 95.9 | 95.8 | $7.03 \times 10^6$ |
| Yolov5s-Bi-FPN | 15.5 | 89.1 | 87.6 | 85.35 | 88.7 | $7.03 \times 10^6$ |
| Yolov5s-SE | 14.8 | 94.9 | 95.1 | 95.2 | 95.6 | $7.23 \times 10^6$ |
| Yolov5s-CBAM | 14.8 | 97.2 | 96.5 | 96.3 | 96.6 | $7.23 \times 10^6$ |

The second experiment's models demonstrated superior performance compared to the first experiment's models. In the third experiment, Yolov7 was used due to its larger network layers and feature pyramids, resulting in longer parameter optimization time and a larger number of parameters. Yolov7 is renowned for its fast and accurate real-time object detection capabilities. The study evaluated the performance of Yolov7 compared to previous models like Yolov5s and Yolov5s-CBAM. The experiment results revealed that Yolov7 had lower performance compared to Yolov5s. Conversely, Yolov5s-CBAM outperformed all the models and demonstrated good performance. Yolov7's detection tasks are often conservative, leading to potential missed targets with lower confidence levels. This results in a slightly lower recall and mAP, as depicted in Table 3. Moreover, Yolov7 has a larger model size than Yolov5s and Yolov5s-CBAM. In conclusion, the experiment highlights that Yolov5s and Yolov5s-CBAM are more favorable choices over Yolov7 in terms of performance and model size. Yolov5s-CBAM, in particular, stands out as the best-performing model among all the evaluated options.

**Table 3.** Results of Yolov5s, Yolov5s-CBAM, and Yolov7.

| Model | Size (MB) | Precision | Recall | mAP@0.5 | F1-Score |
| --- | --- | --- | --- | --- | --- |
| Yolov5s | 14.4 | 96.2 | 96.9 | 95.9 | 95.8 |
| Yolov5s-CBAM | 14.8 | 97.2 | 96.5 | 96.3 | 96.6 |
| Yolov7 | 71.3 | 92.1 | 88.8 | 89.45 | 94.1 |

The model performance is verified by comparing the intersection over union with the highest-performing model in the entire experiment. To do this, we select Yolov5s, Yolov5s-CBAM, and Yolov7 in online inference. The Yolov7 model was chosen to compare its high-parameter performance with low-parameter, well-performed models.

$$IoU = \frac{\text{Area of Intersection}}{(\text{Ground Truth Area} + \text{Predicted Box Area} - \text{Area of Intersection})} \tag{11}$$

The intersection over the union can be explained by dividing the area of intersection between the predicted bounding box and the ground truth bounding box by the combined area of both boxes. Since it represents areas or numbers of pixels, the *IoU* can be expressed in Equation (11). As shown in Figure 10, Yolov5s-CBAM exhibits remarkable accuracy, closely aligning with the ground truth as shown in Figure 10a and covering approximately 98 percent of the ground truth area as shown in Figure 10b. The Yolov5s shows a slight improvement in closeness to the ground truth but still leaves a substantial uncovered area compared to Yolov5s-CBAM of *IoU* = 0.5:0.95, as visible in Figure 10c. Meanwhile, Yolov7 has an even more uncovered area, as illustrated in Figure 10d. The results from Figure 8 confirm that Yolov5s-CBAM offers superior performance in terms of accurately localizing objects compared to Yolov5s and Yolov7.

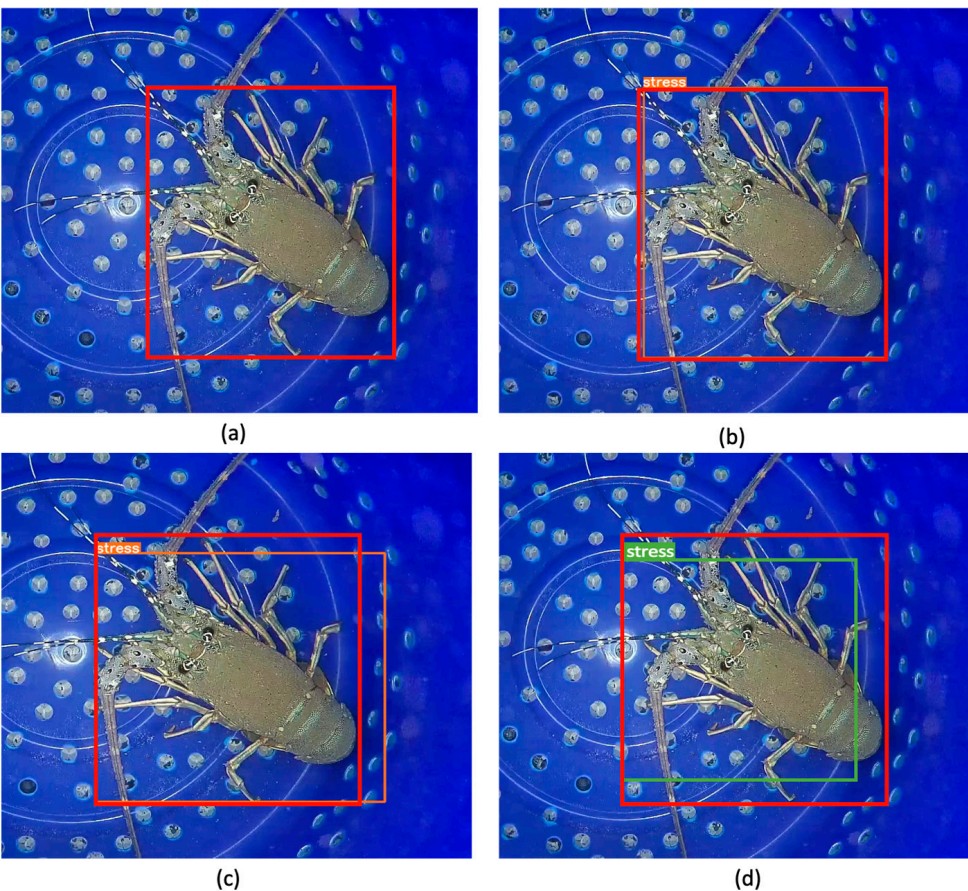

**Figure 10.** Comparison between ground truth (**a**), Yolov5s-CBAM (**b**), Yolov5s (**c**), and Yolov7 (**d**).

The position coordinate prediction loss and confidence prediction loss of the training dataset are represented, respectively, by Train/Box_loss and Train/Obj_loss. The validation dataset's position coordinates prediction loss and confidence prediction loss are represented, respectively, by the variables Val/Box_loss and Val/Obj_loss. The loss value rapidly declines in the first stage before gradually stabilizing, which indicates that the model converges with time, as shown in Figure 11.

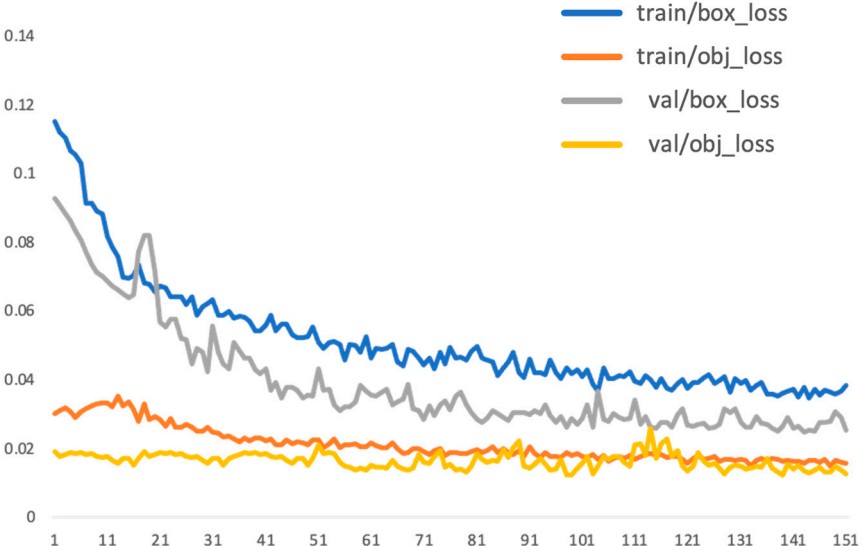

**Figure 11.** Train and eval loss curves.

The *x*-axis of the curve represents recall (R), and the *y*-axis represents precision (P) as shown in Figure 12. As we can see, the recall of the classes gradually increased while precision decreased. This is because the model was more focused on detecting as many lobsters as possible, even if it meant that some of the detections were not accurate.

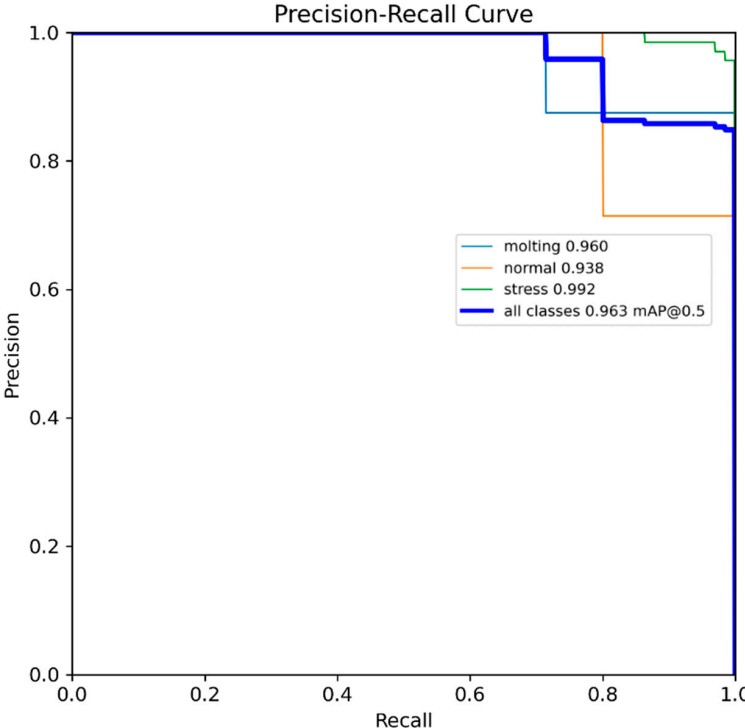

**Figure 12.** Precision and recall plot.

To evaluate the DeepLabCut tool's performance, we conducted a comparison between predicted and ground truth pixels. The goal is to evaluate the tool's effectiveness at detection. To initiate the test, we selected 10 different lobsters molted at different angles to ensure the model's robustness. To compare actual pixel positions with predicted ones, we manually calculated the absolute values of two keypoint positions for ground truth pixels in each lobster image, as illustrated in Table 4. While the predicted results are close to ground truth in most cases, we did identify two instances of misprediction in lobster 7. To conform to this, we conducted a trial involving 50 lobsters in an online setting. The study aimed to determine if false alarms occurred even when lobsters displayed stress postures. Remarkably, we observed no instances of false alarms during the trial, and the tool accurately detected molting lobsters on the online inference.

**Table 4.** Comparison between ground truth pixels and predicted pixels.

| No. of Lobster | GT Pixel of x | GT Pixel of y | Predicted Pixel of x | Predicted Pixel of y |
|---|---|---|---|---|
| Lobster 1 | 1416 | 233 | 1416.9818 | 233.15489 |
| Lobster 2 | 1254 | 389 | 1254.26277 | 389.868271 |
| Lobster 3 | 1412 | 594 | 1414.0636 | 593.483347 |
| Lobster 4 | 1269 | 630 | 1269.84922 | 630.760071 |
| Lobster 5 | 1237 | 361 | 1237.95861 | 361.441697 |
| Lobster 6 | 1244 | 369 | 1245.27473 | 370.97373 |
| Lobster 7 | 1001 | 667 | 1001.21166 | 700.347138 |
| Lobster 8 | 1190 | 336 | 1190.4577 | 336.008819 |
| Lobster 9 | 420 | 1255 | 420.716797 | 1255.80385 |
| Lobster 10 | 669 | 1005 | 669.498929 | 1005.50592 |

## 4. Conclusions

This study utilizes AI and robotics to analyze and detect lobster molting. Using an optimized Yolov5s algorithm and the DeepLabCut tool, we successfully detected all six molting phases, enhancing our understanding of this crucial process. This integrated approach offers promising prospects for efficient and accurate lobster molt detection through evaluation metrics and comparative studies. This study's limitations arise from its inability to adapt to various factors like low lighting, background color, camera types, and the minimal number of lobsters used for determining the molting period. Future plans involve expanding the study to include over 1000 lobsters and conducting domain adaptation to improve its robustness and applicability.

**Author Contributions:** Conceptualization, B.N. and C.-M.L.; methodology, B.N. and C.-M.L.; software, B.N. and R.L.; validation, B.N., V.-D.T. and C.-M.L.; formal analysis, B.N. and C.-M.L.; investigation, C.-M.L.; resources, R.L.; data curation, R.L.; writing—original draft preparation, B.N. and C.-M.L.; writing—review and editing, B.N. and C.-M.L.; visualization, V.-D.T. and R.L.; supervision, C.-M.L.; project administration, B.N., C.-M.L. and V.-D.T.; funding acquisition, C.-M.L. and V.-D.T. All authors have read and agreed to the published version of the manuscript.

**Funding:** This research received no external funding.

**Data Availability Statement:** The data generated during and analyzed during the current study are available from the corresponding author upon reasonable request.

**Conflicts of Interest:** The authors declare no conflict of interest.

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
