# Peer review of "Advanced Robotic System with Keypoint Extraction and YOLOv5 Object Detection Algorithm for Precise Livestock Monitoring"

_fishes, doi:10.3390/fishes8100524_

Round 1
Reviewer 1 Report
This study presents an autonomous monitoring system that uses a target detection and classification network to accurately detect and classify the molting stage of lobsters. This research presents a classification of lobster molting stage into six distinct stages: s1 (normal), s2 (stress), s3-s5 (molt), and s6 (exoskeleton). A dataset of 1000 photographs of lobsters is used for training and testing. The study is interesting, but a comprehensive and meticulous revision of the paper is required.
1. Key Contribution section should be should be streamlined and optimized. For example, refrain from include sentences such as ‘Lobsters are important animals that are used for food.’ that are not pertinent to the topic of key contribution at hand.
2. Section 2 should be combined with Section 1.
3. Lines 75-76: In order to substantiate the assertion of the potential of AI in aquaculture, it's better to investigate additional aspects beyond the provided case study of fish behavior detection and two-species classification detection. Specifically, it is suggested that feeding behavior detection and fish disease detection be included as complementary examples.
4. Lines 81-86: The logical coherence of the argument is not readily apparent; therefore, these sentences need to be revised and refined.
5. Lines 102-104: The temporal range is excessively broad and may lead to reader confusion. It is recommended that the term "recently" be replaced with the specific year "2016".
6. Lines 107-113: The logical coherence of the text is perplexing, and there appears to be a lack of correlation between the sentences. The first sentence be repositioned prior to line 114.
7. Section 3.1: Further elaboration is required about the comprehensive depiction of the data, e.g., encompassing the explicit delineation of labeling categories as well as the quantification of images.
8. Line 196: Problems with formula formatting, harmonize formula formatting
9. Lines 184-186: Is the process of including the CBAM attention mechanism module into the backbone network of YOLO v5s distinct from the one taken by the authors mentioned in quote 17 on line 109, who enhanced the backbone network of YOLO v5s by integrating the CBAM attention mechanism module into it? Upon examination of the primary source, it is evident that the subsequent piece underwent identical modifications as the original article. Hence, the conclusion presented in line 417 lacks validity.
10. Section 4.4: The study uses the F1 score as a performance index to evaluate the effectiveness of the model, considering both precision and recall. However, it is worth considering the use of mAP as an evaluation index, as it provides a more comprehensive assessment of global metrics.
11. Line 328: The different YOLO v5 models i.e. YOLO v5s, YOLO v5m and YOLO v5l need to be briefly introduced in Section 3
12. Line 341: It is necessary to provide a concise explanation for selecting the Bi-FPN and the SE as points of comparison with the CBAM.
13. Figure 7: The confidence of posture detection is transferred to DeepLabCut when it reaches its maximum value, which contradicts the statement made in line 300 regarding the initiation of keypoint detection after achieving majority of the time.
14. Lines 304-306: The YOLO v5s-CBAM model is utilized for the purpose of re-importing in order to detect the state of the ExoSkeleton. Is this model trained in isolation? If not, does it directly detect ExoSkeleton at initial detection?
15. Based on the key point depicted in Figure 8, it is important to conduct observations from a lateral perspective. Is it feasible to achieve a side-view effect in practical observation by utilizing the observation apparatus shown in Figure 6? Can the molting state of a lobster be determined by measuring the European distance between the blue and orange dots? What are the criteria employed for categorizing the Molting and ExoSkeleton states of lobsters, taking into account variations in viewing angles and the sizes of the lobsters?
16. Section 4.4: The absence of DeepLabCut based keypoint detection results and their analysis is notable. The lack of evidence to support the claim that no false alarms were detected in the 30-50 lobster tests (Lines 406-410). It is suggested that more information be included in the article to address this deficiency.
17. Figure 10 did not produce any categorization results for the state S6 (ExoSkeleton). S6 is characterized as a phase in lines 20-21 and described as a posture in line 180. Is S6 classified as a posture, a state or an independent shell? The authors should provide further explanation and present the results of the categorization tests.
18. A number of figures inside the text lack proper figure captions, and the arrangement and labeling of those figures appears to be haphazard. For example, figure 2 should be in Section 3.2; two figure 8, etc.
Extensive editing of English language required
Author Response
Dear Reviewer,
We sincerely thank you for your humble reply to our manuscript
As per your comment, we answer all the questions in both the document's responses to the reviewer and the manuscript.

Reviewer 2 Report
- The data collection section should mention any processing done on the images as well as labeling done for dataset creation.
- The pointers that are mentioned in the conclusion are the contributions of the work. The paper would benefit from moving these to the end of the introduction to provide the reader with an overview of what to expect.
- Authors have presented evidence for their choice of model with sufficient justification.
- Authors have mentioned that only 1000 images are collected and out of which only 800 images are used for training which would cause overfitting. The authors have not mentioned the training details except for the epochs.
In summary, the paper would benefit from restructuring to improve the understanding as well as the flow of the paper.
Author Response

(The authors gave the same response as above.)

Reviewer 3 Report
The article describes work on an autonomous monitoring systems to assist farmers in managing lobsters. Concretely, molting is monitored, i.e., the process where the lobsters shed their exoskeleton. An autonomous robot and computer vision based on deep learning is used.
Regarding the application case, I do not have any expertise. But it is well described and the presentation sounds very reasonable and the demand for the system seems very plausible. With respect to the intelligent systems side, especially the deep learning methods, I can confidently state that the work is very well done. There are no great new methodological contributions in that area, but the adaptation to the application and the non-trivial combination of the methods is thoroughly carried out. This includes the system and methods design, the experiments, and the results that are very convincing.
Author Response

(The authors gave the same response as above.)

Round 2
Reviewer 1 Report
1. It is necessary to modify the captions of figures as they often lack completeness and accuracy.
2. Line 343: The YOLOv5s model did not demonstrate superior performance in terms of mAP compared to other models.
3.I strongly suggest that the authors include the limitations of the current study in the conclusion section.
4.It is advisable to get assistance from a proficient English speaker to refine the text, as it exhibits colloquial language.
Accept after minor revision
Author Response

(The authors gave the same response as above.)
